# Cancer-Associated Fibroblast Diversity Shapes Tumor Metabolism in Pancreatic Cancer

**DOI:** 10.3390/cancers15010061

**Published:** 2022-12-22

**Authors:** Raphaël Peiffer, Yasmine Boumahd, Charlotte Gullo, Rebekah Crake, Elisabeth Letellier, Akeila Bellahcène, Olivier Peulen

**Affiliations:** 1Metastasis Research Laboratory, GIGA-Cancer, University of Liège, 4000 Liège, Belgium; 2Laboratory of Tumor and Development Biology, GIGA-Cancer, University of Liège, 4000 Liège, Belgium; 3Molecular Disease Mechanisms Group, University of Luxembourg, Campus Belval, 4367 Esch-sur-Alzette, Luxembourg

**Keywords:** pancreatic cancer, PDAC, cancer-associated fibroblast, CAF, metabolism, paracrine signaling, desmoplasia, hypoxia, acidosis

## Abstract

**Simple Summary:**

Cancer-associated fibroblasts (CAFs) represent an important stromal cell population of pancreatic cancer in which multiple CAF subtypes have been identified. CAFs engage in a bidirectional crosstalk with cancer cells, continuously adapting their metabolism to external factors, such as chemotherapy. In this review, we summarize recently identified CAF subtypes in pancreatic cancer and discuss how CAFs shape cancer cell metabolism through several mechanisms, notably metabolite exchange, paracrine signaling, desmoplasia/hypoxia and acidosis.

**Abstract:**

Despite extensive research, the 5-year survival rate of pancreatic cancer (PDAC) patients remains at only 9%. Patients often show poor treatment response, due partly to a highly complex tumor microenvironment (TME). Cancer-associated fibroblast (CAF) heterogeneity is characteristic of the pancreatic TME, where several CAF subpopulations have been identified, such as myofibroblastic CAFs (myCAFs), inflammatory CAFs (iCAFs), and antigen presenting CAFs (apCAFs). In PDAC, cancer cells continuously adapt their metabolism (metabolic switch) to environmental changes in pH, oxygenation, and nutrient availability. Recent advances show that these environmental alterations are all heavily driven by stromal CAFs. CAFs and cancer cells exchange cytokines and metabolites, engaging in a tight bidirectional crosstalk, which promotes tumor aggressiveness and allows constant adaptation to external stress, such as chemotherapy. In this review, we summarize CAF diversity and CAF-mediated metabolic rewiring, in a PDAC-specific context. First, we recapitulate the most recently identified CAF subtypes, focusing on the cell of origin, activation mechanism, species-dependent markers, and functions. Next, we describe in detail the metabolic crosstalk between CAFs and tumor cells. Additionally, we elucidate how CAF-driven paracrine signaling, desmoplasia, and acidosis orchestrate cancer cell metabolism. Finally, we highlight how the CAF/cancer cell crosstalk could pave the way for new therapeutic strategies.

## 1. Introduction

Among the multiple hallmarks of pancreatic ductal adenocarcinoma (PDAC), its extensive desmoplastic tumor microenvironment (TME) is regarded by many as crucial for PDAC aggressiveness and prognosis as the PDAC stroma is highly heterogeneous and comprises more than 90% of the tumor volume [1]. The pancreatic TME is constantly changing in composition and adapting to disease progression and external stimuli [2]. On the one hand, PDAC stroma consists of cellular components, including pancreatic stellate cells, fibroblasts, endothelial cells, and immune cells [3]. On the other hand, acellular components are abundantly present, such as collagens, glycoproteins, proteoglycans, and secreted factors [4]. Understanding the composition of the stromal compartment and deciphering the crosstalk between stromal and tumor cells in PDAC has evolved into an important axis of cancer research and drug development. In this review, we first summarize cancer-associated fibroblast (CAF) heterogeneity in pancreatic cancer, focusing on cells of origin, activator signaling, markers, and function. In the second part, we assess how CAFs orchestrate cancer cell metabolism, either directly via metabolite exchange or indirectly via paracrine signaling, extracellular matrix (ECM) production and acidosis. Finally, we discuss the clinical relevance of targeting CAFs by recapitulating how recently proposed drug candidates aim to disrupt CAF/cancer cell crosstalk.

## 2. CAF Heterogeneity in Pancreatic Cancer

Within pancreatic cancer stroma, cancer-associated fibroblasts (CAFs) are regarded as substantial actors, and thus significant progress has been made towards CAF subtype characterization in the last decade. CAFs are generally considered as non-neoplastic, due to the absence of mutations in oncogenic driver genes [5]. However, the stromal expansion of CAFs has been associated with *KRAS* mutations in pancreatic cancer cells [6]. The cellular origin of CAFs in PDAC still requires elucidation, since several cell types have been shown to differentiate into CAFs, including pancreatic stellate cells (PSCs) [7,8,9], resident fibroblasts [10], and adipose-derived mesenchymal stem cells (AD-MSCs) [11]. However, due to the lack of fibroblast-specific markers, tracing the origin of CAFs using genetically engineered mouse models (GEMMs) is challenging, and results must be interpreted cautiously [12]. Accordingly, one such attempt to link different cells of origin, such as PSCs, to distinct CAF subpopulations using PDAC mouse models generated surprising results, whereby PSCs only evolved into a minority of differentiated CAFs [9]. In another study, healthy fibroblasts were discriminated, based on *GLI1* and *HOXB6* expression and upon lineage tracing; both populations only partially contributed to the PDAC CAF landscape [13]. Altogether, these studies highlight the possibility that different stromal cell types could acquire a defined CAF state in response to a common activating stimulus. Thus, when compared to the initial stromal type, the activator signaling of cancer cells towards stromal cells could, to a greater extent, determine CAF differentiation.

Different models, such as in vitro culture/coculture, patient-derived xenografts (PDX), and GEMMs, are currently used to study CAF heterogeneity in pancreatic cancer. Although in vitro models clearly benefit from their simplicity, they do not always recapitulate every aspect of the TME, whereas GEMMs offer a more representative option for studying the human TME. Some CAF subtypes require paracrine signaling from cancer cells [14], the presence of precise inflammatory stimuli [15] or mechanical ECM properties [16] for their differentiation, which can be hard to recapitulate in vitro. PDX could be considered as a better choice than in vitro culture to investigate the TME. However, when compared to GEMMs, PDX bear drawbacks, such as the partial lack of an immune environment and the progressive replacement of human stroma by murine counterparts [17]. Due to the potential discrepancies across findings generated in vitro, in GEMMs, or on human biopsies, we highlighted the respective model used for CAF subtype characterization in the following sections, especially regarding marker expression.

### 2.1. Myofibroblastic CAFs

Myofibroblasts were first described in a physiological context during wound healing. Upon acute injury, resident fibroblasts are activated through transforming growth factor beta (TGFß) signaling and evolve into myofibroblasts, expressing high levels of alpha smooth muscle actin (alpha-SMA) [18]. Myofibroblasts then synthesize constituents of the ECM and basement membranes to restore tissue homeostasis [19]. Furthermore, myofibroblasts acquire contractile abilities via the accumulation of alpha-SMA bundles and myosin fibers, engaging in the mechano-remodeling of ECM to promote wound closure and scar formation [20]. Once the site of injury is healed, myofibroblasts undergo apoptosis or restore a resting phenotype [18,21]. If the injury persists, a continuous repair response can lead to a chronic wound healing syndrome, known as tissue fibrosis. During tissue fibrosis, epigenetic modifications in myofibroblasts enhance anti-apoptotic signaling, resulting in a hyper-activated state that is associated with increased ECM production [22,23]. A continuous tissue repair response is also typical in genetic injuries, such as cancer. Tissue injury, as a result of constant cancer cell accumulation, initiates a chronic wound healing response, better known as tumor fibrosis or desmoplastic reaction. With desmoplasia, fibroblasts are continuously stimulated by cancer cells and evolve into CAFs [24]. In the context of cancer, the difference between myofibroblasts and CAFs is still not entirely defined, as a naming convention is yet to be decided. Myofibroblasts might be the early definition of a CAF, which would later be further divided into different subpopulations, as described below.

The major CAF subpopulation responsible for desmoplasia are myofibroblastic CAFs (myCAFs) (Figure 1A). In human PDAC, myCAFs are in close proximity to the tumor cells and represent 50% of all CAFs, and are thus the most abundant CAF subpopulation within the PDAC stroma [15,25].

The markers commonly used for general CAF identification include alpha-SMA, fibroblast activation protein (FAP), podoplanin (PDPN), and neuron-glial antigen 2 (NG2). However, these markers lack specificity and must be combined with negative markers to eliminate false positives (e.g., endothelial cells or immune cells) [26]. Moreover, due to increasing CAF heterogeneity, underlined by the presence of multiple CAF subpopulations, general CAF markers alone are not sufficient to distinguish between distinct CAF subpopulations. Lately, single cell RNA sequencing approaches have been increasingly used to identify CAF subtypes. These studies came to the consensus that myCAFs are characterized by high alpha-SMA expression, highlighting their functional similarity to physiological myofibroblasts [27]. However, the expression of alpha-SMA should not be regarded as an individual myCAF marker, especially since healthy fibroblasts in colorectal cancers have been shown to bear comparable alpha-SMA levels to CAFs [28].

To increase relevance and certainty when identifying myCAFs in pancreatic cancer, different model-dependent expression signatures have been associated with myCAFs. In 2017, Öhlund and colleagues [29] compared the transcriptome of human myCAFs to quiescent PSCs in vitro and showed an upregulation of *ACTA2* (alpha-SMA), *CCN2* (CTGF), and *COL1A1*. Additionally, a gene list recapitulating the top 25 myCAF markers was included, and acted as a useful tool for gene set enrichment analysis [29]. Importantly, two follow-up studies by other groups confirmed these markers in KIC mice (*Ptf1a*^Cre/+^; *Kras*^LSL-G12D/+^; *Ink4a*^fl/fl^) and human tumors [30,31]. Two years later, using single cell RNA sequencing on human PDAC samples, Elyada et al. [15] characterized myCAFs by increased expression of *ACTA2*, *TAGLN*, *MMP11*, *MYL9*, *HOPX*, *POSTN*, *TPM1*, and *TPM2* [15]. In 2020, Dominguez et al. [25] explored the CAF landscape of pancreatic tumors in KPP mice (*Pdx1*^cre/+^; *LSL-Kras*^G12D/+^; *Ink4a/Arf*^flox/flox^) and human biopsies, identifying a *LRRC15*-positive CAFs subtype that overlaps with myCAFs, as they also show increased expression of *ACTA2*, *TAGLN*, and *MMP11* [25]. *LRRC15*-positive CAFs were shown to have repressive effects on CD8^+^ T cells, and are thus an example of myCAFs playing a role in immunosuppression [32].

An increasing number of CAF markers may be more detrimental than beneficial to our understanding of CAFs. Finding a common characteristic marker among all distinct studies should be a major priority of present research. Even though a recent consensus agreed that a definitive nomenclature is premature, the principal aspect for CAF categorization should be based on the functional phenotypes rather than on the expression of markers [12].

Mechanisms of CAF activation have been thoroughly studied over the recent decades, where multiple cytokines have been shown to activate PSCs (e.g., PDGF, TGFß, IL1) [33]. TGFß is a well-known fibrosis-inducing factor that has been previously well studied in CAF activation [34,35]. TGFß family ligands bind to type 2 TGFß receptor (TGFBR2), which then recruits and phosphorylates TGFBR1 (also known as ALK5) [36]. The resulting heterotetrameric receptor complex then initiates downstream signaling. Canonical signaling is mediated by SMAD2/3 phosphorylation and SMAD4 recruitment. The phosphorylated SMAD complex can translocate to the nucleus and induce target gene expression, due to the presence of SMAD binding elements in promotor regions [37,38]. Non-canonical TGFß signaling, on the other hand, can be independent of SMADs and involves PI3K, ERK, JNK, and RHOA pathways [39]. In PDAC, TGFß signaling has been associated with CAF activation, as TGFß induces alpha-SMA expression and collagen deposition, most probably through a combination of canonical and non-canonical pathways [8,40]. Multiple studies, using GEMMs and human biopsies, link myCAF differentiation and activity to TGFß signaling [15,25,29,32], suggesting that TGFß signaling is crucial for the myCAF subtype. However, it is important to note that several other stimuli have been shown to induce myCAF differentiation in PDAC.

Hedgehog (HH) ligands, such as sonic hedgehog protein (SHH), are known to be abnormally expressed in PDAC [41]. HH ligands have been shown to activate CAFs through interaction with the canonical receptor patched (PTCH1), leading to intracellular signaling via smoothened (SMO) and GLI proteins [42,43]. HH-mediated CAF activation in PDAC leads to stromal expansion and desmoplasia [14,44]. Additionally, HH signaling has also recently been associated with myCAF differentiation [45].

The modifications of mechanical properties of the ECM (e.g., stiffness) have also been shown to stimulate ECM secretion by CAFs. Intriguingly, when CAFs reside in a physiological soft matrix they tend to produce physiological amounts of ECM. However, when CAFs are cultured in a pathological stiff matrix, they are stimulated to produce altered ECM [46]. Accordingly, myCAF-mediated desmoplasia (altered ECM) might act as a positive feed-forward loop on myCAF activity.

The myCAFs have been shown to arise as early as low-grade intraductal papillary mucinous neoplasms (IPMNs), the most common cystic neoplasm and a precursor to PDAC, highlighting the potential importance of myCAF presence during preneoplastic progression [47].

### 2.2. Inflammatory CAFs

Whereas myCAFs are generally characterized by high alpha-SMA expression, low alpha-SMA expressing CAFs have also been identified in PDAC. These low alpha-SMA expressing CAFs are generally referred to as inflammatory CAFs (iCAFs) [29] (Figure 1B). Whereas myCAFs are responsible for ECM production, iCAFs are characterized by immunoregulatory and inflammatory functions, notably via the expression of immunosuppressive ligands (e.g., CXCL12) [12,29,48]. CXCL12 is known for its repulsive effect on T cells recruitment in the TME [49,50] and has been proposed as a target for immunotherapy [51]. Additionally, CXCL12 can act on tumor cells via CXCR4, promoting tumor cell proliferation and dissemination [52]. The iCAFs also show increased expression of IL6 [15,25,29,30]. Previous studies have extensively demonstrated the tumor-promoting effects of IL6-mediated STAT3 signaling. Accordingly, CAF-derived IL6 increases tumor cell proliferation, epithelial–mesenchymal transition (EMT) and metastasis in PDAC [53,54]. An in vitro coculture model involving pancreatic tumor organoids and PSCs, has shown that iCAFs are present at further distance from tumor cells than myCAFs, which reside proximally to tumor cells [29]. This spatial discrimination between myCAF- and iCAF-like CAF subtypes, along with additional clusters, was recently confirmed by spatial transcriptomics of pancreatic tumors. Furthermore, the spatial distribution between CAF subpopulations in pancreatic tumors seemed to be much more complex and nuanced than initially suggested by in vitro coculture [55].

The presence of iCAFs in pancreatic cancer has now been confirmed by several groups, where different signatures (in addition to low alpha-SMA expression) have been identified; all indicating an inflammatory phenotype. During transwell coculture with tumor organoids, iCAFs were characterized by the expression of *IL6*, *IL11*, and *LIF*. A gene list of the top 25 upregulated genes in iCAFs during transwell coculture was published [29]. Similarly, another group described an FB1 fibroblast population in KIC mice, accumulating only at late stages of the disease, which was characterized by the expression of *Il6*, *Cxcl12*, and *Ptn*. This FB1 fibroblast subtype significantly overlaps with iCAFs [30]. Single cell RNA sequencing studies on GEMMs have described larger signatures and also validated previously proposed iCAF signature genes, namely *Il6* and *Cxcl12* [15,25]. Importantly, the expression of *IL6*, *CCL2*, and *LIF* has also been associated with iCAFs in human biopsies [25].

The iCAF differentiation is driven by IL1-induced LIF expression. LIF acts in an autocrine manner to induce STAT3 phosphorylation via the GP130 signaling complex [56,57]. The discrimination between myCAFs and iCAFs in vivo has been explained by a TGFß-mediated downregulation of an IL1 receptor (IL1R) [58]. Fibroblasts located in proximity to tumor cells are thought to evolve into myCAFs, due to the fact that TGFß induces myCAF differentiation and downregulates IL1R, impairing potential iCAF formation. Fibroblasts that are located at a distance from tumor cells are under less influence of TGFß and have been shown to differentiate into iCAFs via IL1 signaling [56]. Recently, increased *TP63* expression was shown to be correlated with high IL1 secretion in squamous subtype pancreatic cancer cells, favoring iCAF differentiation [48]. Additionally, IL1ß—a member of the IL1 family—is thought to be an important regulator of iCAFs. Indeed, while IL1ß was undetectable in PDAC organoids and cell lines in vitro [59,60], it was abundantly present in inflammasome-positive human and mouse tumor cell compartments in vivo. Further investigation found that CAFs are indeed dependent on tumor-derived IL1ß for the production and secretion of immunosuppressive cytokines [61].

The exact timing of iCAFs appearance during tumor initiation and progression remains controversial and seems to be species and context dependent. Whereas studies on human samples show that iCAFs are undetectable during preneoplastic disease stages (i.e., IPMN) and arise only at late disease stages [25,47], other studies on GEMMs clearly reveal the presence of distinct fibroblast populations throughout disease progression and even in a healthy pancreas [25,30].

In general, CAF subtypes should not be considered as fixed. Distinct CAF subpopulations are thought to interconvert and bear important plasticity. Although knowledge in this field is still emerging, preliminary data showed that murine CAFs can interconvert between myCAFs and iCAFs through the alteration of TGFß and IL1 signaling [56]. CAF plasticity is an important PDAC feature to investigate, as it allows constant remodeling and adaptation of the tumor stroma to therapeutic agents.

### 2.3. Other CAF Subtypes

Although myCAFs and iCAFs represent the main CAF subpopulations in the PDAC stroma, other minor subtypes have emerged over the last years.

Antigen-presenting CAFs (apCAFs) are characterized by the expression of CD74, MHC-II, and their ability to present antigens to CD4^+^ T cells (Figure 1C). Since apCAFs seem to lack molecules needed for the stimulation of T cell proliferation, authors hypothesize that apCAFs could act as bait to deactivate CD4^+^ T cells, and thus apCAFs may contribute to immune suppression in PDAC [15]. Mechanistically, apCAFs are thought to be activated via IFN gamma and STAT1 [15]. Although myCAFs and iCAFs have both been shown to arise from PSCs (among others), apCAFs seem to be different. In a KPP mouse model, apCAFs are clustered with mesothelial cells (MCs) from a healthy pancreas, suggesting that instead of PSCs, apCAFs likely originate from MCs [25].

In 2021, metabolic CAFs (meCAFs) were described for the first time in human PDAC (Figure 1D). The meCAFs are specific to patients with loose ECM (low desmoplasia), where meCAFs are suggested to be the most abundant CAF subtype. The meCAFs show high glycolytic activity and are characterized by the expression of *PLA2G2A*, while their differentiation is potentially CREB3L1-dependent. Intriguingly, meCAF abundance negatively correlates with survival but has a positive impact on PD-1-targeted immunotherapy response in PDAC patients. Increased response to immunotherapy in patients with abundant meCAFs potentially arises from the fact that these patients are characterized by loose ECM, thus favoring immune cell infiltration. Additionally, authors suggest a direct crosstalk between meCAFs and T cells [16].

Another recently described CAF subpopulation are meflin^+^ CAFs (Figure 1E). Meflin is a vitamin D-responsive cell surface protein first described as a mesenchymal stromal cell (MSC) marker [62]. Meflin^+^ CAFs bear low alpha-SMA expression and correlate with favorable prognosis in PDAC patients and mouse models. Meflin^+^ CAFs show tumor-restraining properties by inhibiting structural remodeling and crosslinking of collagens, thus favoring a less aggressive TME [63]. Meflin^+^ CAFs underline the importance of defining CAF subpopulations, as not all CAFs should be regarded as tumor promoting.

### 2.4. Stromal Heterogeneity and CAF Subtypes in Pancreatic Cancer

Intertumoral heterogeneity is a well-known hallmark of pancreatic cancer and is regarded as an ongoing burden for therapy development [1]. Tumor microdissection in combination with transcriptomic analysis allowed the classification of PDAC patients based on tumor cell expression data. As an example, Collisson et al. [64] proposed “classical”, “quasi-mesenchymal”, and “exocrine-like” subtypes, whereas Bailey et al. [65] divided PDAC patients into “squamous”, “pancreatic progenitor”, “immunogenic”, and “ADEX” subtypes. Although these studies only consider the tumor cell compartment, it is also important to elucidate a stromal heterogeneity between PDAC patients and the potential repercussions on CAF subtypes.

Moffitt et al. [66] classified the PDAC stroma into two subtypes, “normal” and “activated”. Patients from the “activated” subtype showed a significantly worse median survival when compared to patients belonging to the “normal” subtype. The “normal” stroma was characterized by high *ACTA2* expression, suggesting a potential enrichment in myCAFs, whereas the “activated” stroma showed increased chemokine expression (*CCL13*, *CCL18*), suggesting an inflammatory response and a potential enrichment in iCAFs [66].

In 2021, another study proposed three distinct stromal transcriptional subtypes (S1–S3) for pancreatic cancer, based on gene expression data of microdissected human tumors. S1 subtype, enriched in genes related to development and differentiation, was associated with better prognosis compared to S2 and S3 subtypes, which were enriched in antigen-presenting and macromolecule-modifying related genes, respectively [67]. When applying gene expression signatures from Elyada et al. [15], apCAFs correlated significantly with the S3 subtype. Although statistically non-significant, iCAFs trended towards a correlation with S1, whereas myCAFs seemed to correlate with S2 [67]. In conclusion, the stromal classification of PDAC patients could give valuable indications on the CAF subtypes that populate the TME and guide therapy decisions towards the most effective option.

## 3. CAFs and Tumor Metabolism

During tumor progression, cancer cells are in continuous need for energetic fuel and building blocks to support their superior proliferation rate. Metabolic rewiring describes the mechanisms of how cancer cells adapt their metabolism to meet their bioenergetic demands [68]. The concept that cancer cells have a distinct metabolism from healthy cells was first described by Otto Warburg. In 1920, Warburg published a landmark paper, reporting that cancer cells in aerobic conditions take up excessive amounts of glucose, from which a majority is catabolized into lactate [69]. This observation marked the beginning of what is referred to today as the ‘Warburg effect’, also known as aerobic glycolysis.

However, cancer cell metabolism is far more complex than what Warburg initially described. Aerobic glycolysis not only supplies cancer cells with ATP, it also generates glycolytic intermediates that fuel anabolic pathways necessary to supply cancer cells with non-essential amino acids and nucleotides [70]. Additionally, glucose and glutamine fuel the tricarboxylic acid (TCA) cycle and mitochondrial respiration in cancer cells [71]. Both have been shown to be crucial for pancreatic cancer progression and metastasis formation [70,72,73,74].

It is important to highlight that the metabolic goal of proliferating cancer cells is dramatically different from healthy cells. Whereas non-cancerous cells use glucose and amino acids to fuel their TCA cycle for ATP production, proliferating cancer cells exploit these same pathways, not only to produce ATP but also to generate increased amounts of biosynthetic precursors to meet their high demand in anabolic processes [75]. Excellent reviews deciphering the metabolism in pancreatic cancer cells have been previously published [76,77].

The metabolic rewiring of cancer cells is partly dependent on mutations in oncogenic driver genes, such as MYC and KRAS [78,79]. However, in a tumor, cancer cells are engaged in a tight crosstalk with the TME, exposing cancer cell metabolism to constant extracellular stimuli. Here, we describe how CAFs shape cancer cell metabolism in PDAC, either directly via metabolite exchange or indirectly via paracrine signaling, ECM production, and acidosis.

### 3.1. Direct Effects of CAFs on Cancer Cell Metabolism via Metabolite Exchange

Within the pancreatic TME, CAFs can promote tumor progression by sustaining the metabolic demand of cancer cells via metabolite exchange (Figure 2). In comparison to healthy fibroblasts, CAFs show increased expression of the lactate transporter MCT4, but do not rely on MCT4 for their survival. In fact, CAFs bear increased glycolytic activity and lactate secretion due to elevated *HIF1A* expression. Authors hypothesize that excessive lactate fuels cancer cells and promotes tumor progression [80]. Moreover, tumor-cell-derived TGFß increases aerobic glycolysis in CAFs while impairing oxidative phosphorylation by downregulating *IDH3A*, resulting in high lactate production [81]. Such metabolic crosstalk between CAFs and cancer cells was initially described as a ‘reverse Warburg effect’, during which CAFs show high rates of aerobic glycolysis, fueling adjacent cancer cells with energy-rich metabolites, such as lactate and pyruvate [82].

Exosomes secreted by CAFs can significantly reprogram cancer cell metabolism. In fact, CAF-derived exosomes carry amino acids, TCA cycle intermediates, and lipids. Once taken up by cancer cells, they have been shown to inhibit mitochondrial respiration while promoting proliferation via increased glycolysis and glutamine-dependent reductive carboxylation [83]. The reductive carboxylation of glutamine describes the conversion of glutamine into alpha-ketoglutarate; alpha-ketoglutarate is then further metabolized into isocitrate, which finally generates acetyl-CoA. Increased reductive carboxylation has been associated with cancer cell proliferation in multiple cancer types [84,85,86]. Intriguingly, in contrast to macropinocytosis, exosomal delivery of metabolites to cancer cells is independent of *KRAS* mutations [83]. The macropinocytosis of extracellular proteins was described previously in PDAC as an endocytic process, exploited by *KRAS* mutant cancer cells to fuel their central carbon metabolism with amino acids [87].

Other than through exosomes, CAFs can stimulate branched-chain amino acid (BCAA) metabolism in cancer cells by upregulating branched-chain alpha-ketoacid (BCKA) production. CAFs secrete BCKA, which is taken up by cancer cells and further oxidized to fuel the TCA cycle. Intriguingly, cancer cell-derived TGFß stimulates BCKA synthesis in CAFs by inducing ECM internalization and *BCAT1* expression. BCAT1 is a known catalyzer of BCAAs transamination into BCKA [88].

Alanine, a non-essential amino acid, can outcompete glucose and glutamine to fuel the TCA cycle of cancer cells. Autophagy in PSCs can be stimulated by cancer cells via soluble factors. Increased autophagic flux in PCSs generates excessive amounts of alanine, an alternative carbon source delivered to cancer cells [89].

A CAF subtype specific to PDAC patients with loose ECM, known as meCAFs, are thought to produce metabolic intermediates that fuel oxidative metabolism in cancer cells, promoting PDAC progression [16]. Precise metabolites secreted by meCAFs still need to be identified, although the increased glycolytic activity evident in meCAFs suggests pyruvate and lactate are likely players.

Although most of the metabolite trafficking between CAFs and cancer cells in PDAC seems to be related to amino acid metabolism, lipid-derived metabolites can also be exchanged. For example, PSC-derived CAFs secrete important amounts of lysophosphatidylcholines (LPCs) compared to their healthy counterparts. The extracellular enzyme autotaxin, secreted by cancer cells and CAFs, then converts LPCs into the wound-healing mediator lysophosphatidic acid (LPA). LPA promotes proliferation, migration, and AKT activation in pancreatic cancer cells [90].

### 3.2. Indirect Effects of CAFs on Cancer Cell Metabolism

#### 3.2.1. Paracrine Signaling

In addition to metabolite exchange, CAFs can modulate cancer cell metabolism via paracrine signaling, notably by growth factor and cytokine release (Figure 3A).

A hallmark of PDAC, is the presence of *KRAS* activation mutations in over 90% of tumors [91]. Paracrine mediators, secreted by CAFs or PSCs, were described to modulate the cancer cell metabolome in a KRAS-like manner. In response to stromal cues, cancer cells underwent important epigenetic changes (i.e., increased histone acetylation), leading to MYC activation. Subsequently, cancer cells showed increased glucose consumption and lactate production. Since these metabolic changes overlap with oncogenic KRAS effects, authors considered Ras-inducing factors, such as CTGF or HGF, as key actors [92]. It was indeed confirmed later that PSCs can stimulate glycolysis in pancreatic cancer cells via HGF. PSC-derived HGF binds to its receptor c-MET on PDAC cells, leading to nuclear translocation of YAP and HIF1A stabilization. HIF1A signaling increases the expression of stemness markers and hexokinase 2, resulting in increased glycolysis and lactate production [93].

Interestingly, *KRAS* mutant cancer cells hijack CAFs to enhance oncogenic KRAS signaling and adapt their own metabolism in a reciprocal manner. *KRAS* mutations in tumor cells increases the pool of cytokines secreted (GM-CSF, G-CSF, SHH), and, therefore, are exposed to CAFs in the TME. Canonical HH signaling in CAFs upregulates ECM production, as well as the secretion of IGF1 and GAS6. IGF1 and GAS6 can then reciprocally affect tumor cells, via the IGF1R/AXL-AKT axis. Since *KRAS* mutations can lead to impaired mitochondrial respiration [72], this sophisticated reciprocal signaling node restores mitochondrial respiration in *KRAS* mutant tumor cells via SHH, IGF1R/AXL, and AKT [94].

Focal adhesion kinase (FAK) activity in CAFs can also regulate tumor cell metabolism in a paracrine manner. In fact, FAK depletion was shown to increase CAF-derived CCL6 and CCL12 secretion. Subsequently, CCL6 and CCL12 enhance glycolysis in tumor cells via CCR1/CCR2-mediated protein kinase A (PKA) activation [95].

#### 3.2.2. ECM Production and Hypoxia

During tumorigenesis and tumor progression, CAFs proliferate and generate excessive amounts of ECM (Figure 3B). This process is also called tumor fibrosis, or desmoplasia, and is considered a hallmark of pancreatic cancer [1]. The matrisome is defined as the ensemble of proteins that are part of, or associated with, the ECM. Matrisome composition is tissue dependent and can undergo important modifications during tumorigenesis [96]. In PDAC, the matrisome is highly fibrotic, consisting predominantly of fibrillar collagens (COL1A1, COL1A2, COL3A1, COL6A3) and glycoproteins (fibrillin-1, fibronectin, fibrinogens, periostin). Secreted factors, such as S100 family members, only make up a small proportion of the PDAC matrisome [97]. Excessive ECM production and matrix contraction are both mediated by myCAFs and have been shown to increase interstitial fluid pressure (IFP). Increased IFP is observed in many solid tumors and has been associated with inefficient drug uptake [98]. Another result of excessive ECM production is the establishment of a hypoxic TME. In line with extensive desmoplasia, a highly hypoxic TME is indeed evident in PDAC tumors [99]. ECM production and desmoplasia are known to have multiple effects on tumor progression, such as facilitating invasion and metastasis formation [100]. In this review, we focus mainly on the effects of ECM production on cancer cell metabolism, which is predominantly driven by ECM-induced hypovascularization and hypoxia.

The direct cause for hypoxia is insufficient vasculature, being unable to supply all tissue areas with oxygen. Accordingly, in a patient study, a majority of pancreatic tumor samples were hypovascularized [101]. Whereas desmoplasia is a major cause for hypovascularization [102], the presence of antiangiogenic factors within the PDAC stroma also contributes to impaired vasculature, thereby favoring hypoxia [103]. Additionally, hypoxia acts as positive feedback loop on PSC activation, as hypoxia increases ECM production by PSCs [104]. Although PSCs were shown to stimulate endothelial cell proliferation via VEGF secretion in vitro, it was shown in tumors that PSCs significantly contribute to hypoxia by stimulating cancer cells to produce endostatin, a potent anti-angiogenic molecule, through an MMP-dependent cleavage mechanism [105]. Moreover, hypoxia leads to the increased secretion of SHH by cancer cells, which, in turn, stimulates myCAF differentiation [106]. Thereby, PSCs participate in a vicious cycle, during which activated PSCs favor hypovascularization via increased desmoplasia, which, in turn, differentiates PSCs into myCAFs and promotes ECM production. Additionally, hypoxia can influence the secretion of specific ECM components. Lumican, a proteoglycan, is secreted by PSCs and was previously associated with reduced tumor growth and extended survival in PDAC patients [107]. Interestingly, hypoxia induces autophagic degradation of lumican in PSCs via HIF1A. Reduced lumican secretion, in turn, leads to increased cancer cell proliferation and aggressiveness [108].

Hypoxia has multiple other effects on tumor biology, such as increased metastasis formation [109] and the establishment of an immunosuppressive TME [110]. Cancer cells that reside in hypoxic areas of the tumor undergo an important metabolic rewiring. The master regulator for metabolic adaptation to hypoxia is the transcription factor hypoxia-induced factor 1 alpha (HIF1A). When oxygen is available, HIF hydroxylases (e.g., PHDs) hydroxylate HIF1A on proline residues, subsequently leading to proteasomal degradation by the von Hippel Lindau (pVHL) ubiquitin ligase complex. Since PHD activity is dependent on oxygen, HIF1A hydroxylation decreases when oxygen levels drop during hypoxia. Hypoxia, therefore, leads to HIF1A stabilization, and upon dimerization with HIF1-ß family members, HIF1A translocates into the nucleus to initiate target gene transcription [111]. Cancer cells exposed to hypoxia, therefore, show high HIF1A activity. Additionally, HIF1A is overexpressed in PDAC patients and is associated with poor prognosis [112].

HIF1A rewires cancer cell metabolism in hypoxic PDAC, not only to sustain ATP levels but also, predominantly, to limit reactive oxygen species (ROS) production, since mitochondria under hypoxia were shown to produce excessive amounts of ROS [113,114]. Accordingly, HIF1A orchestrates the switch from mitochondrial oxidative metabolism to glycolysis and lactate production, notably via increased expression of PDK1, LDHA, and PKM2 [115,116]. Moreover, HIF1A was shown to increase autophagy in pancreatic cancer cells, enhancing their migratory capacity [117]. Finally, cancer cell adaptations mediated by HIF1A, can also confer gemcitabine resistance. HIF1A upregulates ABCG2 in an ERK1/2-dependent manner, increasing gemcitabine resistance via drug efflux in PDAC [118].

Cancer cells were also shown to adapt their metabolism under desmoplastic conditions, independent of oxygen levels and HIF1A. In fact, nutrient-deprived PDAC cells were shown to take up collagen I and IV in order to supply the cancer cell with proline via PRODH1-mediated collagen breakdown. Thus, upon the conversion of proline into glutamate, collagen-derived proline acts as an efficient fuel for TCA cycle metabolism, promoting PDAC growth [119].

#### 3.2.3. Tumor Acidosis

The extracellular pH (pH_e_) within a tumor can undergo dramatic changes upon tumor progression. Low pH_e_, generally referred to as tumor acidosis, is often associated with hypoxia [120]. CAFs contribute to decreased pH_e_ via multiple mechanisms (Figure 3C). First, as described above, CAFs favor glycolysis and lactate secretion via increased MCT4 expression and PDAC cell-derived TGFß signaling [80,81]. Lactate secretion plays an important role in pH_e_ regulation, since MCT4 is a lactate/H^+^ symporter [121], generating an important H^+^ efflux from CAFs and leading to acidic pH_e_.

Pancreatic CAFs were shown to have important amino acid metabolism and TCA cycle activity in order to supply cancer cells with metabolic intermediates [83,122]. Importantly, the reductive carboxylation of glutamine and TCA cycle dehydrogenases are significant sources of CO_2_, which subsequently generate HCO_3_^-^ and H^+^ ions. The hydration of CO_2_ into HCO_3_^−^/H^+^ is mediated by carbonic anhydrases (CAs) [123], which are overexpressed in pancreatic cancer cells and CAFs [124]. Additionally, hypoxia directly contributes to pH_e_ acidification via increased CA activity [125]. Therefore, increased amino acid metabolism and TCA cycle activity in pancreatic CAFs significantly contribute to tumor acidosis.

The impact of tumor acidosis on cancer cell metabolism has been extensively studied in previous pieces of work and is well documented [120]. However, PDAC-specific findings are limited, and further research is still needed.

It was recently suggested that acidosis-adapted PDAC cells upregulate YAP signaling and undergo a metabolic shift towards the pentose phosphate pathway. Additionally, this metabolic adaptation in cancer cells was associated with increased invasion [126].

Also, glycolytic tumor cells in low pH_e_ conditions preferably discharge metabolic waste, such as lactate, to adjacent cells via connexin-43 channels. Connexin-43 channels reveal a novel metabolic exchange route, whereby low proliferating PDAC cells in acidic/hypoxic conditions transfer lactate through a cytoplasm syncytium towards high proliferative PDAC cells in normoxic regions [127].

## 4. Clinical Strategies to Target CAF/Cancer Cell Metabolic Crosstalk in PDAC

There is a desperate need for new therapeutic strategies in PDAC, since most patients develop resistance towards current chemotherapeutic agents, such as FOLFIRINOX, gemcitabine, and nab-paclitaxel [128]. Targeting the tumor stroma or interfering with the crosstalk between cancer cells, immune cells, and CAFs has been recently identified as a novel therapeutic approach. Accordingly, multiple CAF-targeting molecules have been developed in recent years [129,130,131].

Targeting ECM deposition to impair tumor metabolism is the first potential strategy. Losartan and PEGPH20 both target collagen deposition by myCAFs and improve drug delivery in pre-clinical models [132,133]. Although combining PEGPH20 with nab-paclitaxel/gemcitabine did not improve overall survival in metastatic patients [134], losartan in combination with FOLFIRINOX showed promising results in locally advanced PDAC [135]. Other than improving drug delivery, ECM-targeting drugs subsequently re-oxygenize the tumor and increase extracellular pH in the TME, which most likely affects tumor metabolism.

In an opposite approach, other molecules can benefit from tumor hypoxia and acidosis enhancing their effect. For example, gold nanorods are transformed into cell-penetrating particles under low pH_e_. The intravenous administration of nanorods significantly improved radiosensitivity of pancreatic tumors in vivo [136].

As discussed above, TGFß is an important factor in CAF/cancer cell crosstalk. TGFß signaling drives myCAF differentiation and rewires CAF metabolism towards glycolysis, which helps to supply cancer cells with metabolic intermediates [29,81,83]. Galunisertib, a potent antagonist of TGFBR1, in combination with gemcitabine, showed promising results in patients with unresectable PDAC [137]. The therapeutic efficacy of galunisertib is likely a consequence of TGFß-dependent myCAF suppression.

HH signaling also represents an interesting target due to its significant implication in CAF differentiation and metabolic crosstalk with cancer cells. Whereas initial studies showed promising results in a PDAC mouse model [138], follow-up studies were disappointing and clinical trials using two distinct HH inhibitors, IPI-926 and Vismodegib, were prematurely terminated due to the lack of benefits for overall survival [130,139,140]. Importantly, targeting HH in PDAC reduced myCAF abundance but also increased the proportion of iCAFs, favoring an immunosuppressive TME and promoting tumor aggressiveness [45]. In another PDAC study, the depletion of alpha-SMA positive CAFs in a GEMM increased metastasis and reduced survival while promoting an immunosuppressive microenvironment [141].

These studies underline that interfering with individual CAF subtypes can lead to a disproportionate amount of CAF subpopulations in the TME, for example, the balance between myCAF and iCAFs. Although targeting myCAFs (e.g., via HH inhibition) reduces desmoplasia, it also increases tumor aggressiveness [142], potentially by altering the proportion of iCAFs, and thus promoting immunosuppression. CAF plasticity can also contribute to the disproportion of CAF subtypes, as myCAFs and iCAFs have been shown to interconvert [56]. Moving forward, interfering with CAF activity, without affecting the proportions of distinct populations or even targeting different CAF populations simultaneously, might represent the most promising CAF-targeting therapeutic strategies.

## 5. Conclusions

Metabolic crosstalk between CAFs and cancer cells in PDAC represents multiple potential therapeutic targets that are yet to be completely addressed. Moreover, previous clinical trials have highlighted the importance of deciphering CAF heterogeneity and demonstrated the complexity of tumor biology. Combining therapies to simultaneously target distinct tumor compartments represents a therapeutic approach with increasing potential.

## Figures and Tables

**Figure 1 cancers-15-00061-f001:**
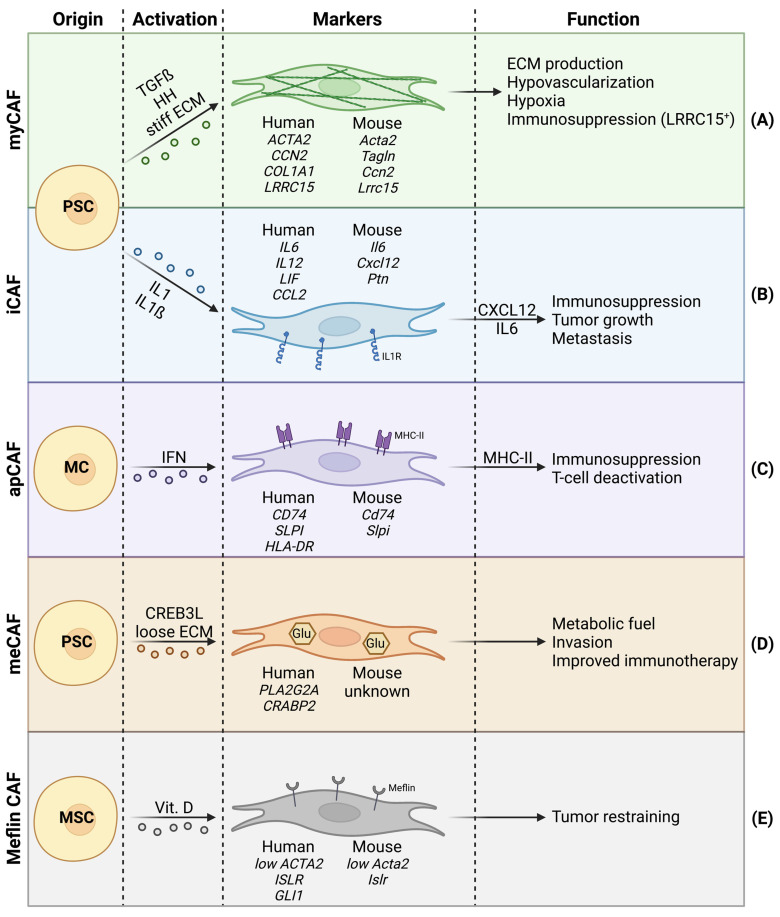
Cancer-associated fibroblast (CAF) heterogeneity in PDAC, with focus on cell of origin, activator signals, species-dependent markers, and functions. (**A**) Myofibroblastic CAFs with high alpha-SMA expression. (**B**) Inflammatory CAFs showing abundant IL1 receptors. (**C**) Antigen-presenting CAFs that express MHC-II. (**D**) Metabolic CAFs bearing increased glycolysis. (**E**) Meflin^+^ CAFs with tumor-restraining properties.

**Figure 2 cancers-15-00061-f002:**
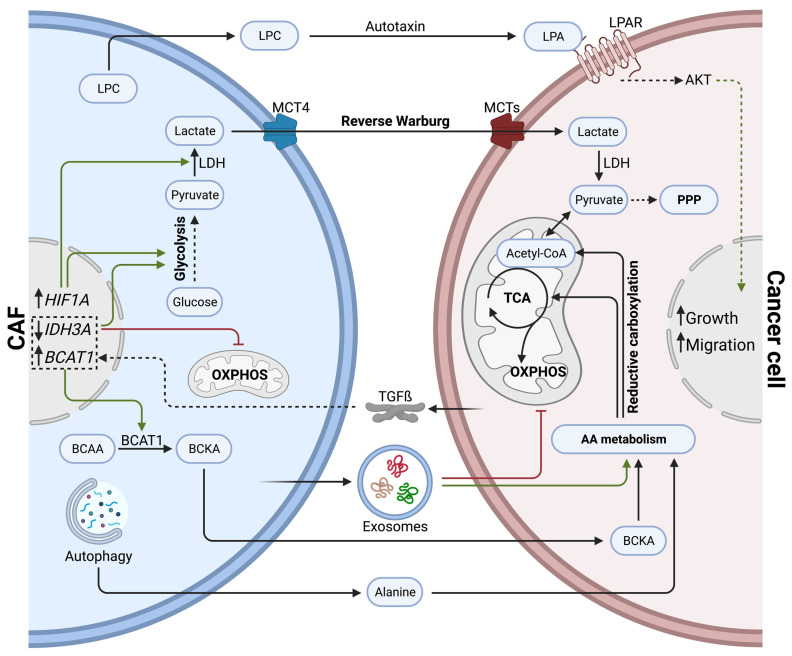
Direct effects of cancer-associated fibroblasts on cancer cell metabolism via metabolite exchange in PDAC. CAFs influence cancer cell metabolism by secreting lactate (reverse Warburg effect), amino acids, lipids, and exosomes. Cancer cells shift towards amino acid metabolism, TCA cycle, and pentose phosphate pathways to support anabolic processes and growth.

**Figure 3 cancers-15-00061-f003:**
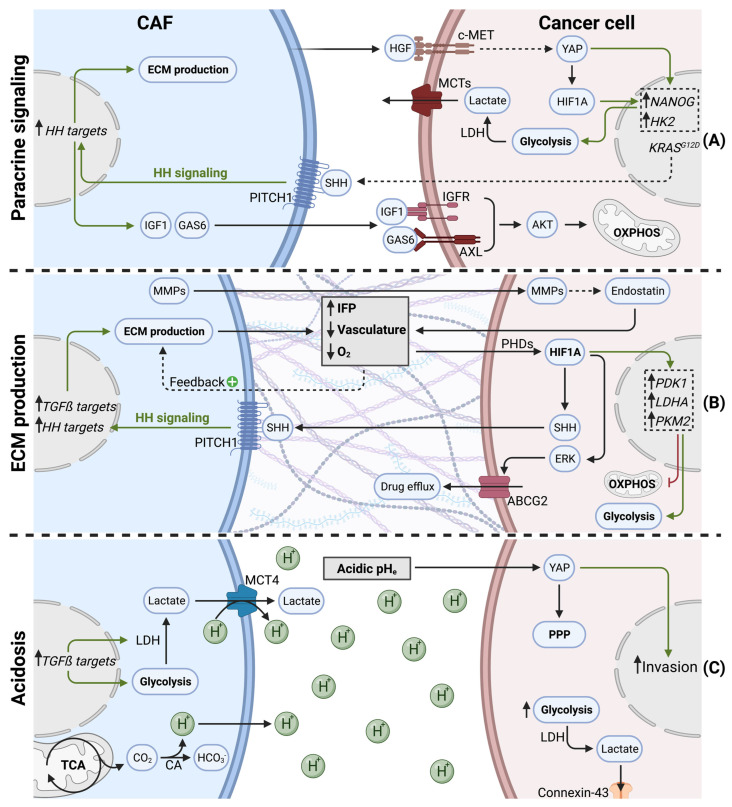
Indirect effects of cancer-associated fibroblasts on cancer cell metabolism in PDAC. (**A**) CAFs secrete cytokines and growth factors to orchestrate cancer cell metabolism. (**B**) CAFs produce ECM (desmoplasia), leading to hypovascularization and hypoxia. Desmoplasia and hypoxia stabilize HIF1A, shifting cancer cell metabolism towards glycolysis and promoting drug efflux. (**C**) CAFs favor acidic pH_e_ (tumor acidosis) via increased H^+^ secretion. Acidosis promotes cancer cell invasion, glycolysis, and metabolic waste trafficking in PDAC.

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
