# Peer review of "Cancer-Associated Fibroblast Diversity Shapes Tumor Metabolism in Pancreatic Cancer"

_cancers, 2022, doi:10.3390/cancers15010061_

Round 1

Reviewer 1 Report

Peiffer et al. submitted a review focusing on subtypes of cancer associated fibroblasts (CAF) in PDAC. The review is well-written and gives a valuable overview of the current literature in the field. Some aspects need to be revised or addressed in more detail.

Major

-          Please add a section about the limitations of in vitro versus different kinds of xenograft models (KPC, PDX model etc.) in recapitulating human CAF populations.

-          Regarding meCAF, please go into further detail about what immunotherapy was efficient and what the underlying mechanism might be.

-          Please comment on potential plasticity in CAF subtypes and the relevance for therapeutic options.

-          The ECM production section is confusing since it covers mostly hypoxia and vasculature. Please make a separate section for hypoxia and another for ECM. The interaction between endothelial cells and CAF should also be mentioned. Please also cover the paradox tumor-promoting versus tumor-restraining effect of ECM (Rhim et al. etc.) and tumor-cell-matrix interaction and their consequences including their role for metastasis.

-          Evaluating clinical strategies of anti-stromal agents needs a more critical approach: Most clinical studies (e.g. SHH-directed agents) have failed to improve patient outcomes. Allude to potential reasons.

-          Stromal heterogeneity and stromal subtypes are not mentioned in the review (Moffit subtypes, Birnbaum et al. etc.). This is an important key to therapy response and crucial for translational approaches. A section about stromal heterogeneity should be added and how that relates to CAF subtypes.

Minor

-          Please change ‘myCAFs have been shown to arise as early as low-grade intraductal papillary 166 mucinous neoplasms (IPMNs), the most common cystic neoplasm and a precursor to 167 PDAC. Highlighting a potential importance of myCAF presence during preneoplastic 168 progression’ (page 4, line 165) to ‘myCAFs have been shown to arise as early as low-grade intraductal papillary 166 mucinous neoplasms (IPMNs), the most common cystic neoplasm and a precursor to 167 PDAC, highlighting a potential importance of myCAF presence during preneoplastic 168 progression’

Reviewer 2 Report

I am delighted to review an excellent and informative paper regarding CAFs diversity and their roles of cancer metabolism in PDAC.

It is thought to be an excellent review paper that is well written and easy to understand using appropriate illustrations. 

So I have nothing to point out for further corrections.

Thank you

Author Response

Authors thank reviewer #2 for his/her nice compliments regarding our manuscript.

Thank you.